# Long-term tracking demonstrates effectiveness of a partnership-led training program to advance the careers of biomedical researchers from underrepresented groups

**Beti Thompson[1], Mary A. O'Connell**  **[2]\*, Karen Peterson[3], Michele Shuster[4], Marilyn Drennan[1], Helena Loest[2], Sarah Holte[1], Julian A. Simon[5], Graciela A. Unguez[4]**

**1** Public Health Sciences, Fred Hutchinson Cancer Research Center, Seattle, Washington, United States of America, **2** Plant and Environmental Sciences, New Mexico State University, Las Cruces, New Mexico, United States of America, **3** Administrative Division, Fred Hutchinson Cancer Research Center, Seattle, Washington, United States of America, **4** Department of Biology, New Mexico State University, Las Cruces, New Mexico, United States of America, **5** Clinical Research and Human Biology, Fred Hutchinson Cancer Research Center, Seattle, Washington, United States of America

\* moconnel@nmsu.edu

**Data Availability Statement:** Third party data underlying Table 1 is available from

## Abstract

The demographic profile of the biomedical workforce in the U.S. does not reflect the population at large, raising concerns that there will be insufficient trained researchers in the future, and the scope of research interests will not be sufficiently broad. To diversify and expand the pool of researchers trained to conduct research on cancer and cancer health disparities, a series of training activities to recruit and train primarily Hispanic students at both the undergraduate and graduate level were developed. The strengths of both a Hispanic Serving Institution and an NIH-designated Comprehensive Cancer Center were leveraged to develop appropriate research training and professional development activities. The career progression of the participants and degree completion rates was tracked, along with persistent interest in biomedical research in general and cancer and cancer health disparities research in particular for these underrepresented individuals. Finally, this report demonstrates that these training activities increased general knowledge about cancer among participants.

## Introduction

The character of the United States (US) population is becoming increasingly diverse, with underrepresented (UR) individuals making up a sizeable segment of the entire population. Despite this increased diversity, there remain inequities in the distribution of UR individuals pursuing careers in biomedical research. The US Census data from 2010 demonstrates the growing populations in the US for UR individuals: 16% Hispanic, 13.6% African American, 1.7% American Indian/Alaskan Native, and 0.4% Native Hawaiian; altogether comprising over 30% of the US population [1]. However, fewer than 11% of the biomedical workforce self-identifies from these UR groups [2]. This dearth of UR individuals in biomedical professions is

https://oia.nmsu.edu/files/2019/01/2018-Quick-Facts-Final.pdf. The authors confirm they did not have any special access to this data. All other underlying data is available within the manuscript and its Supporting Information files.

**Funding:** This work was supported in part by grants from the National Institutes of Health, National Cancer Institute: U56 CA096286 (MOC), U56 CA096288 (BT), U54 CA132383 (MOC, GAU) and U54 CA132381 (BT, JS). The funders had no role in study design, data collection and analysis, decision to publish, or preparation of the manuscript.

**Competing interests:** The authors have declared that no competing interests exist.

exacerbated by an examination of students entering the advanced biomedical field; only 8% of US biomedical doctoral students come from UR groups [2].

In the past three decades, many reports have been published that are critical of the lack of UR individuals in biomedical sciences [3–6]. The inability to achieve workforce diversity is attributed to many factors, but a key issue includes the institutional barriers that prevent students from entering the pipeline [6, 7]. Efforts to change the situation have resulted in very modest gains. As a result, the US is not producing a diverse, inclusive workforce; further, this threatens the nation's international competitive efforts [8]. New strategies and methods must be identified to counter the trends in the demographics of human resources in the biomedical arena to match that of our nation's general population [9].

Promoting a strong and diverse biomedical workforce is a top priority for the National Institutes of Health (NIH). Within the NIH, the Center to Reduce Cancer Health Disparities (CRCHD) at the National Cancer Institute (NCI) is a leader in identifying strategies and mechanisms to increase UR groups in biomedical research [10]. One of several CRCHD programs that has the goal of increasing UR individual participation in the pipeline is the Partnerships to Advance Cancer Health Equities (PACHE) program. PACHE's objective is to "enable institutions serving underserved health disparity populations and underrepresented students and NCI-designated Cancer Centers (CCs) to train scientists from diverse backgrounds in cancer research and to effectively deliver cancer advances to underserved communities" [11, 12]. This report documents efforts made by a specific PACHE program, the Partnership for the Advancement of Cancer Research (PACR) between New Mexico State University (NMSU) and Fred Hutchinson Cancer Research Center (Fred Hutch). Initiated in 2002, the Partnership has implemented training and collaborative research opportunities for primarily NMSU undergraduate and graduate students with the goal to specifically develop diverse leaders in cancer research and cancer health disparities research [13–15].

The Partnership developed an extensive tracking system to document the achievements of the participating undergraduate and graduate students. This system is the foundation of this research, which has three major objectives. First, we demonstrate meeting the goal of developing diverse leaders in cancer research and cancer health disparities by examining the long-term outcomes of our trainees. Then we examine the association between career decisions made by these trainees and their involvement in PACR specific activities. Finally, we show that trainees' dissertation research topics were conducted on cancer and health disparities.

## Methods

### The setting

According to the 2010 census, New Mexico is a minority-dominated state with a population that is 62% UR groups [16]. The 14,289 NMSU students enrolled on the Las Cruces main campus are highly diverse with 61.2% from UR groups: 56.5% identify as Hispanic, 2.1% as Native American, and 2.6% as African American, making NMSU an institution primarily serving underserved health disparity populations and UR students [17]. In the PACHE Partnership, NMSU is the Minority Serving Institution.

Fred Hutch located in Seattle, WA, recognized as an NCI-designated Comprehensive Cancer Center, is the other member of the Partnership. Fred Hutch is seeking ways to involve UR groups in research, education, and outreach activities conducted at the center. Fred Hutch serves more than 5 million residents in their catchment area in Washington state of whom 22.7% are UR groups [18]. Health disparities research is an important goal of Fred Hutch.

## Intervention strategies

A number of intervention activities in PACR have been instituted to assist UR individuals to pursue advanced biomedical careers. An Integrated Training and Evaluation Core (ITREC) oversaw and implemented these research education activities. First, at the undergraduate level, two strategies involved students in research education: Annually, up to six NMSU students were eligible to attend a summer internship at Fred Hutch. This internship provided professional experience working in a faculty-run laboratory with the student pursuing a scientific project on which s/he presented a scientific poster at the end of the internship. In addition, the students experienced social relationships not only with each other, but also with a cohort of interns from around the country who came to Fred Hutch to learn about research, graduate/medical school education, and professional development. The social relationships were encouraged through structured events to which all students were invited, through living in common dormitories, and through common educational experiences such as seminars, workshops, and scientific presentations. Students were strongly encouraged to attend such events and this resulted in the formation of friendships and camaraderie for many students. A second strategy was to place undergraduate and graduate students in laboratories and programs at NMSU conducting cancer research during the academic year. Again, they experienced social relationships as well as an environment that fostered rigorous research and professional development.

At the graduate level, another internship opportunity was available for graduate students; this also was a summer internship at Fred Hutch. Graduate students also benefitted from a Biostatistics Workshop held annually at NMSU and hosted by faculty from NMSU and Fred Hutch.

At Fred Hutch, a health disparities class attracted students who were interested in learning more about health inequities. In the past, Fred Hutch also supported a post-baccalaureate program for students not quite ready for graduate studies. Finally, the Border Experience Program recruited students from Washington state to travel around the Mexico-New Mexico border to experience the challenges of health care in an extremely underserved area. In all cases, individual students benefitted from institutional support, peer social support, and an environment conducive to advanced biomedical research. Such support activities included workshops in writing personal statements for entrance into advanced training, peer review of abstracts and papers, and intensive mentoring. All students met with their mentor on a weekly basis and discussed their scientific project(s), their future academic career, scientific papers to be written as a result of their work, and steps to be taken to pursue career goals. In addition, each student met one or two times during the summer with the academic mentor of the entire program and discussed career goals and opportunities as well as any student questions.

## Tracking participants

Tracking surveys were administered via SurveyMonkey annually in April for at least ten years post-program participation. The surveys were modified slightly to reflect whether the program participant was an undergraduate or graduate student and whether they participated in a summer internship. All of the surveys were electronic questionnaires designed to identify participants' current status, ways in which program participation informed or supported their academic/professional goals, and accomplishments post-program, i.e., degrees completed, scholarships, grants, publications. Print versions of the surveys are provided as supplementary files (S1–S3 Files). Maintaining email contacts for past participants was and remains an ongoing effort, with contacts and information gathered by social media in addition to emails. The Fred Hutch Institutional Review Board (IRB) and NMSU IRB have reviewed and approved

this project annually since the inception of the Partnership in 2002. Three cycles of five-year awards have been completed and results from those cycles are reported here as PACR02/07 (2002 to 2007), PACR07/13 (2007 to 2013), and PACR13/18 (2013 to 2018).

## Dissertation titles

Dissertation titles and abstracts of the PACR trainees who completed their doctoral degree, either during or after participation in PACR, were collected using ProQuest (https://search. proquest.com). Records were searched by trainee name and self-reported degree year to recover dissertation titles and abstracts. The institution awarding the doctoral degree on the ProQuest record was checked against the self-reported institution by the trainee in their tracking survey response. The titles and abstracts were read to determine the research focus of the dissertation. The dissertation was scored as being conducted on (1) cancer research and/or (2) health disparities research, or on (3) biomedical research not including cancer or health disparities, or an (4) other topic., Two independent readers of the abstracts and titles characterized the focus of the dissertation research. These readers met to discuss discrepancies until they attained a consensus.

## Cross-sectional survey

In the interest of comparing the understanding of cancer biology and of health disparities, during the 2015 and 2017 academic years, a cross-sectional survey was conducted of a random sample of trainees who had been involved with the Partnership and a random sample of trainees who had not been involved in the Partnership with the goal of identifying any differences between the two groups in familiarity with and understanding of cancer biology and of health disparities. The sample for this survey included 100 current and prior participants in PACR-sponsored biomedical research education activities (undergraduate and graduate students, and post-docs); and a comparison group comprised of 100 individuals not in PACR, (undergraduates in genetics or cell biology courses, graduate students taking cancer or health related courses and post-docs not working on PACR-funded projects). Of the participants, 48% were undergraduates, 38% were graduate students, and 14% were post-docs, reflecting the overall distribution of trainees in the various PACR funded programs. The participants were sent an email with a link to the SurveyMonkey questionnaire, with one reminder email sent 2 weeks later. A print version of the questionnaire is provided as a supplementary file (S4 File). The Fred Hutch and NMSU IRBs reviewed and approved this project.

## Analysis

Descriptive data are presented for the various surveys. For the cross-sectional survey, a t-test was conducted examining differences between participants and non-participants of the PACR program by percentage correct responses to the health disparities and the cancer biology questions.

## Results

### Demographics of the NMSU student population

As noted previously, NMSU is a designated Minority Serving Institution (MSI), as well as a Hispanic Serving Institution (HSI) because it has a significant percentage of students self-identifying as Hispanic. Table 1 presents the number of undergraduate and graduate students within the NMSU colleges. These are limited to the colleges that have faculty who participate in the Partnership, as the University has some colleges that do not participate in the

**Table 1. Demographics of undergraduate and graduate students in participating colleges at NMSU.**

| College | Undergraduate | Graduate | Total |
|---|---|---|---|
| Agricultural, Consumer & Environmental Sciences | 1,433 (58%[a]) | 203 (23%) | 1,636 (54%) |
| Arts & Sciences | 4,384 (66%) | 801 (30%) | 5,185 (61%) |
| Business | 1,502 (63%) | 299 (44%) | 1,801 (60%) |
| Engineering | 2,160 (54%) | 398 (29%) | 2,558 (50%) |
| Health & Social Services | 1,148 (70%) | 310 (58%) | 1,458 (68%) |

[a]Percentage of students who self-report as Hispanic, African American, Native American, or Pacific Islander; all data from [17].

Partnership. The percent of the students who self-report as members of groups identified as UR in biomedical fields (Hispanic, African American, Native American, Pacific Islander) are also listed.

## Diversity of program participants

Over the past 15 years the proportion of UR individuals participating in Partnership activities has been monitored. The racial and ethnic diversity of the participants is displayed in Table 2. Overall, UR participation for all the funding cycles of the Partnership and across all program activities was 63.6%. The lowest rate of diversity among participants was for the activity that recruited from the Fred Hutch/University of Washington campus, specifically, the Border Experience Program (BEP), with 33% UR participation respectively.

**Table 2. Student participants in NMSU—Fred Hutch Partnership activities (2002–2018).**

| | Total # | # (%) Fem | # Hisp | # NA/PI | # AA | # Other | % URM |
|---|---|---|---|---|---|---|---|
| **PACR02/07** | | | | | | | |
| UG Research projects | 30 | 21 (70.0) | 16 | 7 | 0 | 7 | 76.7 |
| UG Summer interns | 31 | 22 (71.0) | 23 | 3 | 3 | 2 | 93.5 |
| Grad Research projects | 16 | 10 (62.5) | 11 | 2 | 1 | 2 | 87.5 |
| Grad Summer interns | 4 | 3 (75.0) | 3 | 0 | 1 | 0 | 100 |
| **PACR07/13** | | | | | | | |
| UG Research projects | 34 | 23 (67.6) | 10 | 8 | 1 | 15 | 55.9 |
| UG Summer interns | 28 | 23 (82.1) | 23 | 2 | 1 | 2 | 92.8 |
| Grad Research projects | 29 | 9 (31.0) | 11 | 1 | 2 | 15 | 48.2 |
| Grad Summer interns | 11 | 8 (72.7) | 6 | 1 | 1 | 3 | 72.7 |
| Border Experience Program | 27 | 25 (92.6) | 5 | 2 | 2 | 18 | 33.3 |
| Post-baccalaureate | 10 | 9 (90.0) | 9 | 1 | 0 | 0 | 100 |
| **PACR13/18** | | | | | | | |
| UG Research projects | 28 | 24 (85.7) | 17 | 3 | 1 | 7 | 75.0 |
| UG Summer interns | 22 | 15 (68.2) | 11 | 1 | 1 | 9 | 59.0 |
| Grad Research projects | 28 | 19 (67.9) | 13 | 0 | 1 | 14 | 50.0 |
| Grad Summer interns | 13 | 11 (84.6) | 9 | 2 | 1 | 1 | 92.3 |
| Biostatistics workshop | 42 | 23 (54.8) | 17 | 2 | 2 | 21 | 50.0 |

The numbers of students (UG, undergraduate; Grad, graduate) in program activities are listed, sorted by the funding cycle of the partnership. The self-reported demographics of gender (female, F) and race/ethnicity (Hispanic, H; Native American/Pacific Islander, NA/PI; African American, AA) are indicated. Other includes non-Hispanic White, Asian, and individuals who do not report race/ethnicity. The percent female for each program level is provided as well as the overall percent underrepresented minority (URM). Some individuals participated in more than one program activity, and there were eight individuals who participated in more than one funding cycle.

**Table 3. Degree completion of PACR participants.**

|  | Total # | BS | MS | PhD | Other doctoral[1] | % Doctoral |
|---|---|---|---|---|---|---|
| **PACR02/07** |  |  |  |  |  |  |
| UG Research projects | 30 | 27 | 7 | 4 | 5 | 30% |
| UG Summer interns | 31 | 29 | 12 | 8 | 3 | 35% |
| Grad Research projects | 16 | NA | 8 | 11 | 2 | 81% |
| Grad Summer interns | 4 | NA | 1 | 3 | 0 | 75% |
| **PACR07/13** |  |  |  |  |  |  |
| UG Research projects | 34 | 21 | 7 | 2 | 2 | 12% |
| UG Summer interns | 28 | 27 | 8 | 2 | 2 | 14% |
| Grad Research projects | 29 | NA | 16 | 13 | 2 | 52% |
| Grad Summer interns | 11 | NA | 10 | 3 | 2 | 45% |
| Border Experience Program | 29 | NA | 23 | 11 | 0 | 38% |
| Post-baccalaureate | 10 | NA | 3 | 0 | 0 | 0% |
| **PACR13/18** |  |  |  |  |  |  |
| UG Research projects | 28 | 16 | 0 | 0 | 0 | 0% |
| UG Summer interns | 22 | 22 | 0 | 0 | 0 | 0% |
| Grad Research projects | 28 | NA | 16 | 2 | 0 | 7% |
| Grad Summer interns | 13 | NA | 10 | 0 | 0 | 0% |
| Biostatistical Workshop | 42 | NA | 17 | 4 | 0 | 10% |

[1]Other Doctoral: MD, PharmD

### Degree completion for participants

Past participants of these programs have reported outcomes in their academic progression (Table 3). The highest completion rates are for the summer intern undergraduates completing BS degrees, 93% in PACR02/07, 96% in PACR07/13 and 100% in PACR13/18. Similarly, the graduate summer interns have high degree completion rates: 100% earned MS or PhD in PACR02/07 and PACR07/13, while 77% completed MS degrees in PACR13/18. The BS degree completion rates for the undergraduates working on research projects is variable, ranging from 90% in PACR02/07 to 57% in the PACR13/18 cycle. The majority of the graduate students working on research projects in the partnership complete graduate degrees; in the PACR02/07 cycle, 21 graduate degrees were completed by 16 students and 81% of them earned doctoral degrees; in the PACR07/13 cycle, 31 graduate degrees were completed by 29 students and 52% of them earned doctoral degrees; in the PACR13/18 cycle, 18 graduate degrees were completed by 28 graduate students and 7% of them earned doctoral degrees.

### Persistence in biomedical research

The dissertation title and abstracts of past student participants were obtained and used to report on the research topics of those participants. As of Spring 2018, 47 former participants completed their dissertations (Table 4), with 96% earning doctoral degrees in biomedical disciplines. The majority of the dissertations (72%) were on cancer relevant projects, while fewer were on health disparities projects (11%). This persistence of interest in cancer and health disparities research topics by former student trainees indicates that the Partnership is meeting the stated aim to train cancer and health disparities researchers.

### Progression into independent positions in the biomedical workforce

The current education and/or employment status of all participants is surveyed annually. Based on a recent survey, 170 past participants were identified as employed and no longer in

**Table 4. Dissertation research topics of former undergraduate and graduate student participants.**

|  | # PhDs | topic—other | topic—Biomedical | topic–Cancer | topic–Health Disparities |
|---|---|---|---|---|---|
| **PACR02/07** |  |  |  |  |  |
| Undergraduate | 8 | 0 | 8 | 6 | 0 |
| Graduate | 11 | 0 | 11 | 10 | 3 |
| **PACR07/13** |  |  |  |  |  |
| Undergraduate | 2 | 0 | 2 | 1 | 0 |
| Graduate | 22 | 2 | 20 | 13 | 6 |
| **PACR13/18** |  |  |  |  |  |
| Undergraduate | 0 | 0 | 0 | 0 | 0 |
| Graduate | 2 | 0 | 2 | 1 | 1 |
| Total | 45 | 2 | 43 | 32 | 11 |

The number of student participants in each of the three phases of the partnership that completed dissertations are listed. The number of dissertations that reported research on cancer and/or health disparities is also listed.

school. These individuals reported their job titles and places of employment on the survey. Sample job titles associated with our participants included Assistant Professor, Clinical Data Analyst, Policy Fellow, Health Practitioner, and Quality Control Specialist. Close to 80% of the respondents were employed in careers that are either in biomedical science or are related to biomedical science. These employment descriptions were clustered into either health care practice (doctors, nurses, public health workers, etc.); research (post-docs, professors, research scientists, etc.); or data or information science areas. The remaining jobs were categorized as other. Equal numbers of participants worked in research (n = 60) or health care (n = 60), while 13 past participants worked in data science areas, and 37 worked in other fields.

Among the past participants, the number of former trainees who were now in faculty and scientist positions in the biomedical workforce was determined. These individuals would have completed doctoral level degrees and were employed with job titles beyond the post-doctoral level. There were 48 individuals in our tracking database who met this definition. Of those, 16 individuals are in faculty positions and 32 are research scientists or clinicians. The individuals in these independent faculty and scientist positions reflect the diverse population of our participants in that 68.8% of the faculty and 59% of the research scientists identify as UR.

## Impact of participation on cancer knowledge

A cross-sectional survey was conducted of participants and non-participants in the Partnership on both the NMSU and Fred Hutch campuses, first in 2015 and again in 2017. The survey assessed cancer health disparities knowledge (five questions) and cancer biology knowledge (five questions). The average percent correct answers for these two categories of knowledge are presented in Table 5.

The knowledge demonstrated by the correct answers to this survey was outside of any classroom or workshop setting. In both years for each category of knowledge, partnership participants had higher scores for correct answers than non-participants, although only the difference in health disparities knowledge was significant (p = 0.02).

## Discussion

In this work, the outcomes of training programs hosted by a partnership between a cancer center and a minority-serving institution demonstrate positive effects on UR students entering

**Table 5. Comparison of cancer knowledge between participants and non-participants of Partnership training activities.**

| Knowledge Area | 2015 (n = 32)* | | 2017 (n = 57) | |
|---|---|---|---|---|
| | Participant | Non-Participant | Participant | Non-Participant |
| Cancer Health Disparities** | 55.6 | 22.2 | 88.1 | 65.0 |
| Cancer Biology | 59.4 | 33.3 | 49.2 | 44.0 |

*Numbers were too small for significance test.

** For 2017, p value for difference in health disparities was 0.02.

the biomedical pipeline. Throughout the three cycles of Partnership funding, UR students participated in Partnership activities. Further, a high proportion completed markers of successful progress, such as theses and dissertations. A sizeable percentage of students have continued to receive advanced academic degrees, and many are now in academic or research positions.

Similar to other long-term tracking of career outcomes [19], PACR past participants who completed doctoral degrees pursued careers in a wide range of professions, including academia. The degree completion rates for the participants in the PACR program are equal to, if not, higher than the degree completion rates reported for other programs. For example, the career progression of undergraduate participants in a 10-week summer program at Emory University has been reported for cohorts over a 15 year period and approximately 27% of the participants were from underrepresented groups [20]. Of the 800 participants, 28% completed any type of graduate degree and the authors did not report on the graduate degree completion rates for their UR participants. A program designed to increase Latino doctorates in cancer public health research has reported on outcomes of ~100 past participants [21]. In this case participants have earned an MS level degree and are trained for the transition to doctoral degree programs; 43% of participants applied to doctoral programs after the training, and close to 30% were enrolled. In both of these examples, very different training programs were conducted, and different outcome metrics reported. As listed in Table 3, in the first PACR cycle, 39 graduate degrees were earned by 61 undergraduates (64%); in the second PACR cycle, 23 graduate degrees were earned by 62 undergraduates (37%). The graduate degree completion rates for participants in the PACR graduate training programs were high: in the first cycle, 25 graduate degrees were completed by 20 participants (125%); in the second cycle, 83 degrees completed by 79 participants (105%). The varied doctoral degree completion rates between undergraduate and graduate students is expected, as graduate students are much closer on their career paths to this goal. The number of undergraduates and graduate students who complete doctoral degrees, increase as more time expires. For example, the percent of undergraduates who complete doctoral degrees doubles between PACR cycles 1 and 2. Perhaps these trainees face financial or familial barriers to immediately entering and completing graduate degree programs. Further survey studies would be needed to address this hypothesis.

The PACR experience provides many lessons. Importantly, there are difficulties in understanding the impact of a training program on UR individuals in the short term. Students do not necessarily traverse the pipeline in a linear manner. Further, long-term outcomes take time to mature. The analysis reported here, was conducted, after approximately ten years, to ascertain the outcome of career progression. When students drop out of and re-enter the pipeline, final progress may take longer.

Annual tracking to ensure annual contact with current and former students was key. This continued contact kept survey response rates from students in the range of 75% to 80%, depending on participant group. High survey response rates are likely facilitated by establishing a personal relationship with each UR individual who is in the pipeline.

Although the numbers are very small, it appears that participation in the Partnership has increased knowledge of cancer biology and health disparities. In both cases, students and faculty within the Partnership were more knowledgeable about the topics than non-participants. This may be because the Partnership has a myriad of activities that promote information dissemination about cancer biology and health disparities; thus, individuals within the Partnership are exposed to more information about the topics. Alternatively, participants may become more interested in the topics and seek out more information.

As part of the surveys conducted by the program, participants often provided responses that helped us understand how they perceived the program. A few examples are provided here:

"The training program gave me exposure to border health and Hispanic health disparities research. It also cemented my interest in racial disparities research and outreach."

(Participant in Border Experience Program).

"I found that there were gaps in my knowledge and that simply learning from in-class lectures is not enough. There is a certain amount that you can learn from hands-on experience that is invaluable. I also learned some tips for applying to medical school and a great deal from the research seminars. I would say that my research skills have improved exponentially since coming here."

(Participant in undergraduate summer intern program)

"The [program] helped me meet my short-term employment goals through the guidance and advice offered in applying for jobs post-internship. [Program staff] helped edit my resume and cover letter for two job applications. Knowing that I need to continuously update my resume and make alterations based on the job description that I am applying for will help my short- and long-term employment goals."

(Participant in graduate summer intern program)

These responses by the participants confirm many of our ideas on the importance of the diversity of program activities we provide through our training and professional development programs. These participants are reflecting on the value of the continuous support for professional development as well as the excellent research exposure of the different programs. The identification of writing skills and support by our program participants has been described by other groups as well [22].

Past trainees who are currently in independent roles have begun to mentor and train diverse students. This speaks to the sustainability of the Partnership approach; just as past trainees learned about the importance of involving UR individuals in the pipeline, so they mentor and encourage diverse individuals when they enter academic or research fields.

There are several strong bases for including UR individuals in academic biomedical fields. A more diverse workforce is likely to contribute to reductions in health disparities overall; diverse investigators may pursue research of questions that address health disparities at a variety of levels, including the biologic, social, environmental, ethical, and other topics that impact underserved populations. Diverse investigators will bring relevant cultural perspectives to research questions that may not be apparent to scientists from majority backgrounds. Recruitment of students from diverse ethnic/racial groups will be increased and maintained as they observe similar role models working in the biomedical enterprise. Finally, the visibility of UR

faculty, fellows, and students in doctoral granting universities will provide effective role models to others pursuing advanced degrees in the biomedical field.

## Supporting information

**S1 File. Undergraduate intern tracking survey.**
(PDF)

**S2 File. Graduate intern tracking survey.**
(PDF)

**S3 File. Non-intern tracking survey.**
(PDF)

**S4 File. Cancer knowledge survey.**
(PDF)

**S5 File. Responses to cross-sectional surveys.** This excel file reports the answers to the cancer biology and health disparities knowledge questions used to calculate the data in Table 5, sorted by year conducted: 2015 sheet and 2017 sheet.
(XLSX)

## Acknowledgments

The authors thank their colleagues at NMSU and Fred Hutch who have served as research mentors, given scientific presentations, and generally enriched the training program for the PACR participants. The authors thank the PACR participants who have completed the annual tracking survey each year, allowing us to monitor their career progression.

## Author Contributions

**Conceptualization:** Beti Thompson, Mary A. O'Connell, Julian A. Simon, Graciela A. Unguez.

**Data curation:** Marilyn Drennan, Helena Loest.

**Formal analysis:** Sarah Holte.

**Funding acquisition:** Beti Thompson, Mary A. O'Connell, Julian A. Simon, Graciela A. Unguez.

**Methodology:** Beti Thompson, Michele Shuster, Marilyn Drennan, Helena Loest.

**Project administration:** Mary A. O'Connell, Karen Peterson, Michele Shuster, Julian A. Simon, Graciela A. Unguez.

**Supervision:** Mary A. O'Connell, Karen Peterson, Michele Shuster.

**Writing – original draft:** Beti Thompson, Mary A. O'Connell, Graciela A. Unguez.

**Writing – review & editing:** Beti Thompson, Karen Peterson, Michele Shuster, Julian A. Simon.

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
