## [Decision Letter · Decision Letter 0]

17 Oct 2019

PONE-D-19-21375

Long-term tracking demonstrates effectiveness of a partnership-led training program to advance the careers of biomedical researchers from underrepresented groups

PLOS ONE

Dear Dr. O'Connell,

Thank you for submitting your manuscript to PLOS ONE. After careful consideration, we feel that it has merit but does not fully meet PLOS ONE’s publication criteria as it currently stands. Therefore, we invite you to submit a revised version of the manuscript that addresses the points raised during the review process.

Please revise your manuscript accordingly to both reviewers comments. In particular, grammatical  errors and term consistency.

We would appreciate receiving your revised manuscript by Dec 01 2019 11:59PM. To enhance the reproducibility of your results, we recommend that if applicable you deposit your laboratory protocols in protocols.io, where a protocol can be assigned its own identifier (DOI) such that it can be cited independently in the future. For instructions see: http://journals.plos.org/plosone/s/submission-guidelines#loc-laboratory-protocols

We look forward to receiving your revised manuscript.

Kind regards,

Cesario Bianchi

Academic Editor

PLOS ONE

Journal Requirements:

2. We note that Figure(s) in your submission contain copyrighted images. All PLOS content is published under the Creative Commons Attribution License (CC BY 4.0), which means that the manuscript, images, and Supporting Information files will be freely available online, and any third party is permitted to access, download, copy, distribute, and use these materials in any way, even commercially, with proper attribution. For more information, see our copyright guidelines: http://journals.plos.org/plosone/s/licenses-and-copyright.

1.    You may seek permission from the original copyright holder of Figure(s) to publish the content specifically under the CC BY 4.0 license.

Additional Editor Comments (if provided):

Dear Dr O´Connell:

Thank you for submitting your manuscript that was revised by 2 experts. Before I can make a final decision, I need that you revised the manuscript accordingly the both reviewers comments.

In particular, there are issues with grammar and some terms used (consistency). Please revise the manuscript and answer all questions raised.

Reviewers' comments:

Reviewer's Responses to Questions

**Comments to the Author**

1. Is the manuscript technically sound, and do the data support the conclusions?

Reviewer #1: Partly

Reviewer #2: Yes

2. Has the statistical analysis been performed appropriately and rigorously? 

Reviewer #1: Yes

Reviewer #2: Yes

3. Have the authors made all data underlying the findings in their manuscript fully available?

Reviewer #1: Yes

Reviewer #2: Yes

4. Is the manuscript presented in an intelligible fashion and written in standard English?

Reviewer #1: Yes

Reviewer #2: No

5. Review Comments to the Author

Reviewer #1: Lines 57 – 63

One of several CRCHD programs with the goal of increasing ethnic/racial individual participation in the pipeline is the Partnerships to Advance Cancer Health Equities (PACHE) program which has as its objective to “enable institutions serving underserved health disparity populations and underrepresented students and NCI-designated Cancer Centers (CCs) to train scientists from diverse backgrounds in cancer research and to effectively deliver cancer advances to underserved communities” [11, 12] Make this 2 sentences; it’s a run-on

Lines 71 – 75

Introduction

First, we examine how well the goal of developing diverse leaders in cancer research and cancer health disparities is being met by examining the long-term outcomes of our trainees. Then we examine the association between career decisions made by trainees and involvement in specific activities. Finally, we show how trainees’ dissertations are linked to cancer and health disparities. How do these 3 major research objectives align with the stated goals listed in the abstract?

General Note

Be consistent throughout the paper with use of the terms “racial and ethnic minority”, “ethnic/racial minority”, “minorities”, “underrepresented groups”, “underrepresented populations”, “underrepresented individuals”

Lines 99 – 101

In addition, the students experienced social relationships not only with each other, but also with a cohort of interns from around the country who came to Fred Hutch to learn about research, graduate/medical school education, and professional development. Were these “social relationships” passive or actively and structurally facilitated? If the latter, please describe and discuss the impact on the student

Lines 116 – 118

Such support activities included workshops in writing personal statements for entrance into advanced training, peer review of abstracts and papers, and intensive mentoring. It would be helpful to hear more about this intensive mentoring and what it consisted of

Line 136

Use of “We” in first person seems out of place

Lines 173 – 174

Table 1 presents the number of undergraduate and graduate students within the NMSU colleges that have faculty who participate in the Partnership. Why is mention of “…that have faculty who participate in the Partnership” relevant? This is not clear

Table 2

List what “UG” means under table

Discussion

In general, the manuscript reports “outcomes” not “evidence”, explicit factors or components of their programming which supports the outcomes of their trainees

There is no mention of varied degree completion rates among undergraduate and graduate students outlined in Table 3, i.e., potential factors, either self-reported or hypotheses, that contributed to this and how this could be addressed moving forward

There needs to be more direct evidence provided about how the program actively influenced persistence in biomedical research, specifically in relation to cancer relevant projects

Lines 290 – 292

We consider the graduate degree completion rates for participants in our undergraduate training programs to be respectable. Similarly, the graduate degree completion rates for participants in our graduate training programs are also very good: What does “respectable” mean, and how is “very good” being determined in comparison to what?

Lines 297 – 298

Further, long-term outcomes take time to mature. For us, it took approximately ten years to assess career progression. This is not an “assessment” as much as a report on outcomes

Lines 313 – 335

Refer to and leverage information provided here to help support claims made throughout the manuscript related to “evidence”

Lines 340 – 350

This information should have been included in the Introduction

Reviewer #2: This is an interesting study, and I appreciate that it is longitudinal, as those are often not undertaken. Charts may add to understanding for readers as many numbers were reported.

Manuscript includes several of grammatical and punctuation errors. There are also a couple of places for more clarity. "We" is used throughout in places third person is more appropriate (e.g. line 63). Also, "its" is used often and seems awkward. Specific notes are below though not exhaustive.

Lines 58-59 needs commas

Lines 71-75 is confusing, needs rewording for clarity

Line 83 needs "primarily" or "predominately" added after "institution"

Line 86, "forms" to "is"

Line 94, usually "as follows" would be followed by a colon. May also want to add "First..." for the first described strategy to add clarity to paragraph.

Lines 102-103, excessively long. Add "undergraduate and graduate" before "students" on 102 and the clause after colon on 103 is not needed.

Line 126, "i.e." usually off-set by comma or parentheses, not colon.

Lines 141-142, too many "then" in process explanation (also appears elsewhere in manuscript. This could be more descriptive and clear.

Lines 152-156, multiple punctuation styles makes the sentence confusing

Line 195, "Individuals" should not be capitalized

Line 226-227, table title uses capital letter for each word

Lines 228-230, "who" and "which" should actually be "that" in both cases

Line 250 "resultant" is awkward in sentence and also the research scientists should be a "resultant" category

Line 259 ends with comma

Line 273 "have gone on" to "continued"

Line 274 needs a comma after "degrees"

Line 277 should have commas to offset "if not higher"

Line 294 under line number has weird symbol, maybe a track changes mark

Line 298 "us" to "this project" or other similar term--paper is too familiar

Lines 313-329 holds good information and examples, but comments would be more appropriate in results than discussion

Line 342 "on" to "of"

Reference 10, I think "D" is "Disparities" and pages need to be reviewed for proper citation notation

6. PLOS authors have the option to publish the peer review history of their article (what does this mean?). If published, this will include your full peer review and any attached files.

Reviewer #1: Yes: Jabbar R. Bennett

Reviewer #2: Yes: Andrea M Zimmerman

---

## [Author Response · Author response to Decision Letter 0]

30 Oct 2019

We have provided full and complete responses to all of the editorial and reviewer comments in our response to reviewers document attached to this submission

---

## [Editor Report · Decision Letter 1]

15 Nov 2019

Long-term tracking demonstrates effectiveness of a partnership-led training program to advance the careers of biomedical researchers from underrepresented groups

PONE-D-19-21375R1

Dear Dr. O'Connell,

We are pleased to inform you that your manuscript has been judged scientifically suitable for publication and will be formally accepted for publication once it complies with all outstanding technical requirements.

With kind regards,

Cesario Bianchi

Academic Editor

PLOS ONE

Additional Editor Comments (optional):

Dear Dr. O´Connell,

Thank you for carefully revising the manuscript. It is an important work that I am glad to accept for publication in the revised version.
---

## [Editor Report · Acceptance letter]

5 Dec 2019

PONE-D-19-21375R1 

Long-term tracking demonstrates effectiveness of a partnership-led training program to advance the careers of biomedical researchers from underrepresented groups 

Dear Dr. O'Connell:

I am pleased to inform you that your manuscript has been deemed suitable for publication in PLOS ONE. Congratulations! Your manuscript is now with our production department. 

With kind regards,

on behalf of

Dr. Cesario Bianchi 

Academic Editor

PLOS ONE